# Potential of Process-Induced Modification of Potato Starch to Modulate Starch Digestibility and Levels of Resistant Starch Type III

**DOI:** 10.3390/foods14050880

**Published:** 2025-03-04

**Authors:** Moshit Yaskin Harush, Carmit Shani Levi, Uri Lesmes

**Affiliations:** Laboratory of Chemistry of Foods and Bioactives, Department of Biotechnology and Food Engineering, Technion—Israel Institute of Technology, Haifa 3200003, Israel; moshityaskin@campus.technion.ac.il (M.Y.H.); shanilc@bfe.technion.ac.il (C.S.L.)

**Keywords:** starch, in vitro digestion, retrogradation, digestibility of starch, resistant starch, autoclave, high-pressure processing

## Abstract

Starch digestibility and the content of resistant starch (RS) play a crucial role in human health, particularly in relation to glycemic responses, insulin sensitivity, fat oxidation, and satiety. This study investigates the impact of processing methods on potato starch digestibility and RS content, focusing on two modification techniques: autoclaving and high hydrostatic pressure (HHP), followed by retrogradation at different temperatures. The research employs a comprehensive approach to characterize structural changes in starch samples using X-ray diffraction (XRD), attenuated total reflectance–Fourier transform infrared (ATR-FTIR) spectroscopy, and scanning electron microscopy (SEM). In turn, semi-dynamic in vitro digestion experiments based on the INFOGEST protocol were conducted to assess starch digestibility, while RS content was evaluated through enzymatic digestion of the non-RS fraction. SEM, XRD, and FTIR measurements reveal thermal processing appreciably affected starch architectures while HHP had a marginal effect. Further, the FTIR 1045/1022R ratio was found to be correlated with RS content measurements while reducing rapidly digestible starch (RDS). The findings led to the stipulation that thermal processing facilitates amylose leaching and granular disruption. In turn, retrogradation enabled the deposition of the amylose onto the disrupted structures which delineated their subsequent liability to enzymatic digestion. Conversely, HHP had minimal effects on granular architectures and amylose leaching. Overall, this research provides valuable insights for processing starch-based food products with the goal of increasing RS content, which may have significant implications for the food industry and nutritional science.

## 1. Introduction

Carbohydrates are a staple and important nutritional source that may impact human health in various pathways. In this context, potato is the fourth most consumed carbohydrate source in the world and its main macronutrients are embodied in starch [1]. Potato starch, like most starches, is the main plant energy source conserved in complex supramolecular architectures, termed starch granules. These are composed of two key polysaccharides—amylose and amylopectin—interwound in a semi-crystalline structure that contains both amorphous and crystalline regions [2,3,4].These intricate polysaccharide assemblies undergo various physicochemical changes during processing and storage which delineate starch hydrolytic breakdown in the human gastrointestinal tract (GIT) upon ingestion [3,5,6,7].

In fact, starch is typically classified into three types based on its digestibility: [i] rapidly digestible starch (RDS) digested within 20 min; [ii] slowly digestible starch (SDS), which takes 20–120 min to digest; and [iii] resistant starch (RS), which is the fraction of starch that resists digestion in the small intestine and reaches the large intestine where it can be fermented [8]. Numerous studies focus on elucidating the balances between RDS, SDS, and RS in starchy foods and their impact on glycemic responses, health, and well-being [9,10]. The high consumption of RDS-rich foods, such as white bread [9], has been shown to have adverse health effects, namely through the disruption of the body’s blood–glucose balance and tolerance [9,10,11]. On the other hand, SDS, found in most raw cereal starches [12], contributes to regulating glycemic responses [13]. RS is commonly found in amylose-rich plants, such as green bananas, legumes, and cereal grains, and is hardly digested by human enzymes [14]. It has been shown to undergo colonic fermentation and promote bile acid turnover, enabling its classification as a beneficial prebiotic [15]. Thus, RS consumption is advocated for and studies demonstrate it may affect post-prandial blood glucose levels, insulin sensitivity, fat oxidation, and satiety. Also, it may reduce the risk of specific intestinal diseases and non-alcoholic fatty liver disease and lower total cholesterol [7,16,17,18,19,20,21,22,23,24,25,26].

Five RS classifications have been defined: RS1 and RS2 are raw and poorly bioaccessible forms of starch and RS4 and RS5 are chemically modified and molecular complexes of [22,27]. RS3 denotes fractions of processed starch that undergo retrogradation into architectures that are poorly digestible. Today, there is a growing debate about the risks and benefits of processed foods which stimulate studies into the ramifications of food processing on its healthfulness. To this end, food processing has been shown to modulate RDS/SDS/RS balances in real food systems [27]. Thus, there is a need for efforts to harness processing to develop RS-rich foods and improve our ability to deliberately control the balance of RDS/SDS/RS.

A range of methods have been employed to modulate starches and attain desired properties for various industrial applications, including chemical, enzymatic, and physical strategies. Among the different types of modification, chemical modifications of starches are the most common due to their cost-effectiveness and simplicity of application [28]. These processes typically involve substitution, degradation, or cross-linking, with oxidation, esterification, and etherification being the most common reaction pathways used.

However, chemical modifications produce significant byproducts and/or side streams that can pose environmental risks. Consequently, physical modifications are preferred and considered “cleaner” or “greener” methodologies for starch [29,30]. Physical modification methods have gained significant attention due to their environmentally friendly nature, cost-effectiveness, and chemical-free labeling [28,30,31,32,33].

Physical modifications applied to starch typically involve changes to its macromolecular architectures or physical state without altering its chemical [30,31,33]. One such approach is the combined use of commercially viable heat and moisture, e.g., cooking or autoclaving, to physically modulate starch [31,32,34,35,36,37]. Subsequently, the retrogradation of staches may occur and yield resistant starch fractions, namely [22,38,39]. Overall, physical modifications may include heat treatment (dry and/or moisturized), high-pressure processing, ultrasonication, microwaving pulsed electric fields, annealing, and extrusion [32,33,40]. These processes have been shown to alter starch’s physical properties such as granule size, crystallinity, gelatinization temperature or water absorption capacity, and the RDS/SDS/RS ratio and, in a few studies, even to affect starch digestibility.

Since the 1980s, various studies have probed the impact of various unit operations on the level of RS [41]. Such studies have characterized the impact of physical and chemical processes or their combinations on RS content in various food systems [42]. These studies focused on improving starch’s techno-functionality for food applications, e.g., pasting properties, and reported changes in the balances between the fractions of RDS, SDS, and RS of various starches [43]. More recent studies have segregated between processing and food formulation effects, i.e., with some highlighting the impact of processing parameters (time, temperature, etc.) while others have demonstrated the importance of formulation (amylose content, the amylose–amylopectin ratio, water content, etc.). This research explores the impact of thermal and non-thermal processing combined with retrogradation on the digestibility of starch using updated consensus in vitro digestion protocols adapted to the semi-dynamic conditions of improved bio-relevance [44]. The main aim was underpinning the pathways to increase the RS content in starch while tilting the ratio of RS and SDS at the expense of RDS.

## 2. Materials and Methods

### 2.1. Materials

Potato starch was donated by Galam company (Galam Ltd., Kibbutz Maanit, Israel). The following materials were purchased from Sigma-Aldrich (Rehovot, Israel):

Pepstatin-A (CAS# 26305-03-3), Glycodeoxycholic Acid sodium salt (CAS# 16409-34-0), Taurocholic acid, sodium salt hydrate (CAS# 345909-26-4), α-Amylase from *aspergillus oryzae* (86250), Pepsin from porcine gastric mucosa (P7000), and Pancreatic α-amylase (A3176). The activity for the 3 enzymes mentioned above was tested in-house, as detailed [45].

α-Amylase from *Bacillus licheniformis* (A4551), Amyloglucosidase from *Aspergillus Niger* (10113), and Protease from *Streptomyces griseus* (P5147) were used as received within three months of purchase, with activity units determined using the data provided in the TDS.

Simulated saliva and gastric and duodenal fluids were prepared following the INFOGEST protocol [45]. Digestive enzymes for the simulated fluids were used according to in-house determinations of the activity of the purchased enzyme powders, as detailed previously [45]. All chemicals used were of analytical grade and were utilized without further modification. Infrared was used to prepare all solutions. 

### 2.2. Modification of Starch

This study used two processing operations to modify starch after dispersion (10% *w*/*v*) in phosphate buffer (10 mM, pH = 7.0), one used thermal processing in an autoclave and the second used high pressure.

To investigate the impact of starch processing on its digestibility, two physical modification methods were selected based on the previously discussed advantages of physical modification. 

Among the various commonly used physical modification processes, autoclaving was selected as the thermal treatment method due to its effectiveness in significantly increasing the RS content and its impact on digestibility, compared to alternative techniques such as annealing and heat–moisture treatment [37,46,47,48]. To provide a comprehensive understanding of the relationship between thermal processes, the increase in RS content, and its impact on digestibility, a comparative analysis was also performed using HHP, a widely utilized non-thermal method [49].

#### 2.2.1. Starch Modulation via Autoclaving

Controlled moisture–heat treatment was performed in an autoclave (121 °C, 20 min). Autoclaved starch samples were allowed to retrograde for 24 h at 4 °C or 40 °C (to align with practical applications in the food industry) to enable the assessment of the impact of retrogradation temperature. They were then lyophilized, pulverized, and sieved through a 30 mesh. The samples were then kept in sealed desiccators for further analyses and termed A-4 °C and A-40 °C, according to the temperature allotted for retrogradation, respectively. The moisture content of native starch was 14.3 ± 0.1%, whereas the moisture content of the samples subjected to autoclaving, followed by retrogradation and lyophilization, was 7.6 ± 0.8%.

#### 2.2.2. Starch Modulation via High Hydrostatic Pressure (HHP)

This process was used as a non-thermal alternative to autoclaving. Herein, starch suspensions were placed in vacuum-sealed plastic packaging. The packed and sealed suspensions were processed using a high-pressure processor (Stansted Mini Foodlab Model S-FL-085-09-W, Harlow, UK) using pre-set conditions (target pressure 550 MPa and holding time 15 min to simulate commercially viable conditions). The HHP-processed starch samples (named HHP—4 °C) were also allowed to retrograde for 24 h at 4 °C, after which they were lyophilized, pulverized, sieved using a 30 mesh, and stored in sealed desiccators until further analysis. To examine the impact of different processing methods on digestion and RS formation after retrogradation, the HHP process was performed to compare to autoclaving, and the HHP-processed samples were allowed to retrograde at 4 °C just as was performed following autoclaving.

### 2.3. Physiochemical Characterizations 

This study sought to explore the nano-, meso- and micro-properties of starch as they are important to starch’s functionality and digestibility [27,50]. Thus, native and process-modulated potato starch samples were characterized by a battery of structural characterization methods including X-ray diffraction (XRD), FTIR, and scanning electron microscopy (SEM).

#### 2.3.1. X-Ray Diffraction (XRD)

These measurements offer insight into the long-range molecular arrangement and crystallinity–amorphous balances, arising from organized arrays of double helices formed by the amylopectin side chains [51]. In potato starch, a B-type spectral pattern can be observed, which is reflected in a strong peak at the 2θ of 17°, medium strength peaks at the 2θ of 15°, 22°, and 24.0°, and a characteristic peak at about the 2θ of 5.6°. Each XRD measurement was performed on ~200 mg of powdered sample placed on an XRD holder and irradiated at room temperature with Bruker D8 -Advanced (Bruker, Billerica, MA, USA) equipped with Cu Kα radiation (λ = 1.5406 Å) operating with a target voltage of 40 kV and a current of 40 mA. Intensities were recorded in the 3–40° 2θ range with a step size of 0.01°/1 s. Crystallinity was calculated and baseline-corrected using Origin 2018 software. The relative crystallinity (RC), representing the percentage of crystalline material in the samples, was calculated using Equation (1), as outlined by Colussi et al. [52]:RC (%) = (Ac/(Ac + Aa)) × 100 (1)
where Aa represents the amorphous area and Ac represents the crystalline area.

#### 2.3.2. Attenuated Total Reflectance–Fourier Transform Infrared (ATR-FTIR)

FTIR analyses provide insights into short-range crystallinity which augments XRD findings. Starch samples were clamped directly on the ATR-FTIR crystal and 48 scans in the range of 1200–900 cm^−1^ at a resolution of 4 cm^−1^ were recorded (using a Thermo Scientific Nicolet iS50 FTIR spectrometer, Thermo Fisher Scientific, Waltham, MA, USA). The background spectrum of the empty cell was used to correct background effects and was recorded under the same conditions. During measurements, the sample was in contact with the universal diamond ATR top plate and data were collected using Omnic9 software. The GAUSSIAN function was used for deconvolution. Infrared (IR) absorbance values at 1045, 1022, and 995 cm^−1^ were extracted from the spectra after background subtraction, baseline correction, and deconvolution using Origin 2018 software. Intensity measurements were determined from the deconvoluted spectra by measuring the height of the absorbance bands from the baseline.

#### 2.3.3. Scanning Electron Microscopy (SEM)

Starch samples, native or processed, were mounted on double-sided tape and coated with carbon using a thermal evaporation machine (Safematic GmbH, Zizers, Switzerland) for high resolution. The Apreo2 Thermofisher SEM (xT microscope Software v23.1.0) was used at a 15 Kv accelerating voltage. Micrographs were taken at 350×, 2000×, and 3000× magnification, taken with secondary electron beams at high-vacuum EDT, and low-vacuum LVD.

### 2.4. In Vitro Digestibility of Starch

Starch digestion is a critical intersection between foods and human health; thus, we used semi-dynamic in vitro digestion (IVD) experiments following the international consensus INFOGEST standardized method and adjusted to better address starch digestion and offer improved bio-relevance [44,45,53].These IVD experiments simulating adult gastrointestinal digestion were conducted using a dual auto-titration unit (Titrando 902, Metrohm, Herisau, Switzerland) equipped with a continuously stirred bioreactor kept at 37 °C through its double-walled design. The IVD model used TIAMO 2.5 software to generate gastric pH gradients based on the physiological data relevant to food research [54,55].

In brief, a sample of 1 g native or processed potato starch was vortexed with 25 mL phosphate buffer (10 mM, pH = 7.0) for 5 s and then mixed with 25 mL simulated salivary fluid (SSF) (to reach a final starch concentration of 20 mg/mL) containing α-amylase (86250, final concentration of 75 U/mL). The received bolus of 50 mL was vortex for an additional 2 min. Subsequently, the oral bolus was added to 50 mL of simulated gastric fluid (SGF) containing pepsin (2000 U/mL) preheated to 37 °C. The initial pH was set to be 4.5 and was adjusted using auto-computer-controlled titration (0.3 M HCl) prior to the launch of the pH gradient program set in TIAMO software to run for 2 h of gastric digestion. After the gastric phase, gastric effluent was mixed 1:1 *v*:*v* with simulated duodenal fluid (SDF) preheated to 37 °C and containing pancreatic α-amylase (200 U/mL), bile salts (5 mM Sodium glycodeoxycholate and 5 mM Taurocholic acid sodium salt hydrate). The intestinal phase duration was set at 2 h, with a static pH of 6.25 that was kept static using 0.3 M NaOH. During IVD experiments, aliquots of 0.7 mL were collected before and after the oral phase; three gastric effluents at time lapses of 5, 15, and 120 min followed by five intestinal samples collected at 5, 15, 30, 60, and 120 min of the intestinal phase.

All samples were inactivated as follows: The oral and intestinal samples were inactivated by adding 4 volumes of ethanol to each sample. Similarly, gastric effluents were inactivated by using 14 μL PepstatinA that were added to each aliquot. All inactivated samples were stored at −20 °C for further analysis.

### 2.5. Evaluation of Starch Digestion Using Dinitrosalicybe (DNS) Acid Assay

The reducing sugars content of native and processed starch was determined using the DNS method [45,56]. Briefly, 1% (*w*/*v*) DNS reagent consisting of 1 g 3,5-dinitrosalicylic acid, 0.05 g sodium sulfite, and 1 g sodium hydroxide in 100 mL DDW was prepared. The calibration curve R^2^ > 0.9995 was formed using 1% (*w*/*v*) maltose stock, diluted to concentrations of 0–0.5% (*w*/*v*) maltose. Each digestion sample was thawed and centrifuged (11,000× *g*, 10 min) to remove the digestive debris. Next, the samples were diluted (1:3 *v*/*v*) with DDW and 240 μL was vortexed with an equivalent volume of 1% DNS reagent and incubated (95 °C, 10 min) to allow the development of the red-brown color. In turn, 61.4 μL of 40% potassium sodium tartrate solution was added to the sample to stabilize the color. After cooling to room temperature, the samples were loaded into 96-well plates, absorbance at 575 mm was measured with a plate reader (BioTek Synergy H1 microplate reader, Winooski, VT, USA), and values were converted to maltose concentrations based on the calibration curve.

### 2.6. Assessing the Potential RS Content

Based on the literature, RS content was estimated through the enzymatic digestion of the non-RS fraction, enabling the isolation of RS in the sample. This method relied on a modified version of the protocol noted previously [19,57,58].

In brief, a 10 g sample was added to 500 mL phosphate buffer (pH = 6, 10 mM). The solution was then heated to a temperature of 90 °C and supplemented with 1.6 mg of heat-stable α-amylase (A4551, 2000 U/L). The samples were incubated for 30 min at 90 °C under continuous stirring, then cooled on ice down to room temperature. Next, the pH of the solution was adjusted to pH = 4.5 using 2% phosphoric acid, and a sample was heated to 60 °C. Then, 2 mL of Amyloglucosidase (2300 U/L) was added and 2 h were allotted for continued starch digestion. Lastly, pH was adjusted with 5 M NaOH to pH = 7.5 when protease (20 U/L in phosphate buffer, 10 mM) was added to allow the complete overnight breakdown of enzymes under continuous stirring. The day after, the samples were centrifugated (5000× *g*, 16 min, 10 °C), decanted, and washed with 96% ethanol 1:1 *v*/*v* before another round of centrifugation. Sediments were collected and centrifuged (1085× *g*, 15 min, 25 °C) to remove any ethanol leftovers, prior to sample lyophilization, and pulverization for further analysis. The RS yield was calculated from the ratio of the sediment obtained after the digestion process to the total starch subjected to digestion.

## 3. Results and Discussion

The link between starch digestion and its health implications stimulates research into the determinants delineating starch digestibility. Thus, this study explores the ramifications of processing and retrogradation to potato starch physiochemical characteristics, its potential digestibility, and the content of RS.

### 3.1. Characterization of Starch and Processed Starch Samples

First, XRD was deployed as a simple qualitative and comparative measure of the long-range-ordered molecular structures in samples, e.g., the crystalline–amorphous balance, as it is criticized for its accuracy [59]. A summary of the XRD spectral patterns of the samples is given in Figure 1, which affirms the overall semi-crystalline nature of all samples. Native potato starch exhibited a typical spectral pattern with a peak at 2θ of 17°, additional intensity peaks at 2θ of 15°, 22°, and 24.0°, and a characteristic peak at about 5.6° 2θ [60].

The XRD patterns of processed and native potato starch presented in Figure 1 reveal a clear distinction between native and processed starch but did not exhibit noteworthy differences between either of the processed samples. One noticeable difference between the processed samples was noted in the disappearance of the peak at 2θ ≈ 5.6°, which is attributed to a transformation between the B and A polymorphs [5,61]. Further, SEM micrographs provide additional information on the nature and regularity of starch morphology resulting from processing the native potato starch granules (Figure 2). While native potato starch (Figure 2A) demonstrated smooth, oval-shaped granules, both thermally processed (Figure 2B,C) and retrograded starch granules (A-4 °C and A-40 °C) were highly disrupted into a flaky appearance indicative of the occurrence of gelatinization. Interestingly, the HHP-processed samples (Figure 2D) did not register appreciable differences in granular architectures compared to the native potato starch which suggests limited gelatinization occurred.

In turn, FTIR was used to gain a better insight into the physicochemical properties of the starch samples (Figure 3).

First, FTIR measurements (Figure 3A) corroborated that neither autoclaving nor HHP treatments led to the formation of new chemical groups, supporting the notion that processing only modulated starch architectures in the short and long-range orders but not starch chemistry. Noticeable intensity differences in the peaks in the 900–1200 cm^−1^ range indicate changes in molecular mobility attributable to changes in crystallinity. Moreover, the literature indicates that the bond strength ratio at 1022/995R can be used to monitor short-range structure loss. That is, the 1022/995R ratio assesses the loss of the molecular arrangement of starch’s double helices within the granules/crystallites, while the ratio at 1045/1022R unveils the extent of the short-range order [62]. Thus, a higher ratio at 1045/1022R has been suggested to be indicative of a more organized structure [51,63,64]. From the deconvoluted peaks seen in Figure 3B, the ratio between the various wavenumbers indicative of crystalline and amorphous fractions can be explored (Figure 4).

Notably, the ratio 1045/1022R (Figure 4A) indicative of crystallinity revealed that the A-40 °C and A-4 °C samples were more crystalline compared to the native starch, while the HHP-4 °C sample had a lower crystallinity, which resembled that of native starch. When analyzing the peak ratio 1022/995R indicative of amorphous fractions (Figure 4B), a similar trend was observed suggesting a shift in the balance between crystalline and amorphous regions in the A-40 °C and A-4 °C compared to the native starch while HHP was practically unaltered compared to the native starch.

### 3.2. In Vitro Digestibility of Processed Starch

Starch breakdown into smaller bioaccessible saccharides through the human upper GI system was investigated in vitro to monitor degradation into di- and monosaccharides, which were examined using the DNS method [56]. The findings of the bioaccessible reducing sugars in the digesta of processed and unprocessed potato starch are presented in Figure 5 in which two main observations are noticeable. First, all samples were partially degraded in the early stages of the gastric phase, which is mainly attributed to residual salivary α-amylase activity and partial acid hydrolysis that are possible due to the dynamics of gastric acidity upon ingestion (addressed by the semi-dynamic mode of the in vitro digestion model). This concurs with various studies into the intricacies of starch digestion that note that salivary amylase and gastric conditions enable the slight breakdown of starch in the stomach [65,66]. Physiologically, gastric effluents are further broken down by pancreatic amylases during the intestinal phase. Herein, both thermally processed and retrograded samples (A-4 °C and A-40 °C) were more readily digestible than the native starch or the HHP-processed starch. The A-40 °C samples were slightly more digestible than the A-4 °C samples and both were more digestible than HHP-processed or native and unprocessed starch. Since starch breakdown correlates with blood glucose levels, the cumulative starch digestion, i.e., the area under the curve (AUC), was assessed based on the release of bioaccessible carbohydrates. The AUC appears highest for the A-4 °C and A-40 °C samples while the HHP-processed samples had a lower AUC which could suggest a possible lower glycemic response.

Starch may also impact health through the prebiotic fraction of RS, hence further tests investigated the levels of indigestible RS in concurring samples (Figure 6). These analyses confirmed that native starch and HHP starch had the least RS content while both cooked and retrograded samples (A-4 °C and A-40 °C) had noticeable RS content. Interestingly, thermally processed samples allowed to retrograde under an elevated temperature (40 °C) yielded higher RS levels compared to samples retrograded under chilled conditions (4 °C).

To date, various methods have been described in the literature to modify starch structure via thermal, non-thermal, and chemical techniques [48,67,68,69,70]. The outcomes of such processing operations can be affected by differences in process parameters as well as starch’s intrinsic characteristics, e.g., starch type and amylose content. Yet, few studies holistically address the ratio between the various starch fractions, i.e., the RDS/SDS/RS ratio, which is increasingly challenged [66,71,72]. Moreover, a screening tool is needed to facilitate distinguishing different starch samples based on the prediction or estimation of possible changes in RS content. Intriguingly, a qualitative correlation was observed between the quantity of RS produced from processed starch samples and the peak ratio 1045/1022R obtained in FTIR measurements (Figure 7).

Such an observation was also noted by others who tested the effects of autoclaving on purple sweet potatoes [42] or different kinds of oats [73]. This relationship suggests that the 1045/1022R FTIR ratio can be used to estimate alterations in resistant starch content in processed starch products. Conversely, the ratio 1022/995R had no correlation to the produced RS content which is also in agreement with Dorneles et al. [74]

Altogether, the findings herein and the current body of evidence make it possible to stipulate that starch undergoes several architectural changes from its native form through gelatinization and retrogradation down to its digestion in the alimentary canal as illustrated in Figure 8. First, hydration and processing disrupt native granular structures in which amylose is naturally intercalated between the amylopectin architectures (that serve as the granule scaffold). Second, the amylopectin matrix swells and accommodates the leaching out of amylose chains in the process of gelatinization. Third, during storage retrogradation occurs in which the released amylose chains are deposited onto the shrinking amylopectin matrix. The conditions in which retrogradation occurs, i.e., the temperature of storage, delineate the mobility of amylose and the order of its deposits or the formation of amylose complexes. Fourth, the drying of starch can entail various modes of starch depositions onto the granular remnants or intercalation, one type of direct amylose deposition and another of pre-formed amylose complexes that generate less ordered moieties.

To this end, retrogradation temperature facilitates higher macromolecular mobility and the formation of more ordered architectures, as can be monitored by their short-range organization and crystallinity. That is, thermal processes increase the molecular mobility of amylose and amylopectin chains that facilitate amylose leaching from the granules while the high-pressure conditions applied herein seem to have a marginal effect on the native starch granule and the leaching of amylose. Last, the starch architectures are exposed to the aqueous digestive fluids and enzymes whose action is governed by the degree of order of the biopolymer system. This translates into the differences in their susceptibility to degradation underlying starch bioaccessibility. Thus, this work provides evidence to support the notion that processing and retrogradation can be used to physically modify starch structures that divert their digestive fate and augment existing evidence [31,34,72,75].

## 4. Conclusions

Understanding starch digestion can open opportunities to harness processing to modulate starch digestibility and its content of prebiotic RS type 3 [22,72]. This work offers three main insights: First, that starch thermal processing and retrogradation attenuate its digestibility and increase its RS content. Second, increasing retrogradation temperature was found to enable the formation of more ordered starch assemblies which exhibit modulated digestion and higher RS content. This is augmented by the 1045/1022R FTIR ratio metrics that were found to be useful in estimating RS content. Third, the gathered evidence indicates that the HPP processing of starch does not induce gelatinization like thermal processing. However, HPP attenuates starch’s digestibility profile due to what seems to arise from the better preservation of the native granular structures. This may also explain the marginal impact of HPP on the RS content of starch, which may be attributed to limited amylose leaching. The findings emphasize the potential of using processing techniques to modulate starch digestibility, which can lead to strategic choices of processes to control starch bioaccessibility interwound with glycemic responses and possibly healthier outcomes. Future work could deepen insights into the effects of advanced processing and retrogradation on starch digestibility and impact on microbiota trajectories as well as correlate in vitro findings with in vivo data and human trials. Thus, this study will ultimately provide a comprehensive overview of how processing and retrogradation can be harnessed to modulate starch digestive behavior in different consumers and help make informed decisions on starch processing toward healthier outcomes.

## Figures and Tables

**Figure 1 foods-14-00880-f001:**
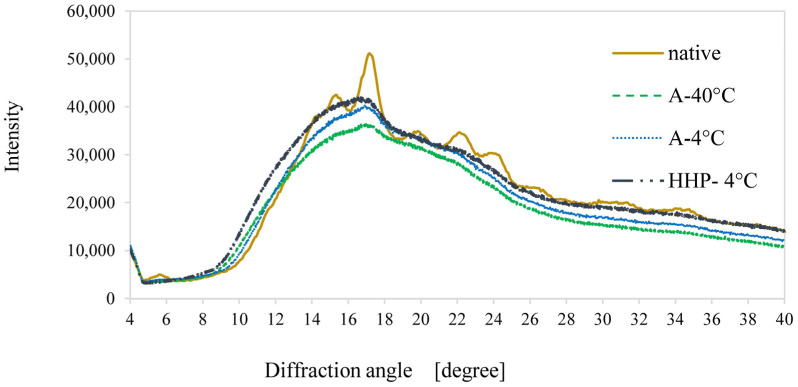
X-ray diffraction (XRD) of native potato starch granules (Native **˗˗˗˗˗**) and retrograded starch after autoclaving and retrogradation at 40 °C (A-40 °C **˗ ˗ ˗ ˗ ˗**), 4 °C (A-4 °C **․․․․․**) or starch treated by HHP followed by 4 °C retrogradation (HPP-4 °C ▬ ▪ ▪ ▬). Patterns with typical B-type polymorph peaks at the 2θ of 17°, moderate intensity peaks at the 2θ of 15°, 22°, and 24°, and a characteristic peak at about 5.6° 2θ. All processed starch samples exhibited a complete disappearance of the 2θ of 5.6°.

**Figure 2 foods-14-00880-f002:**
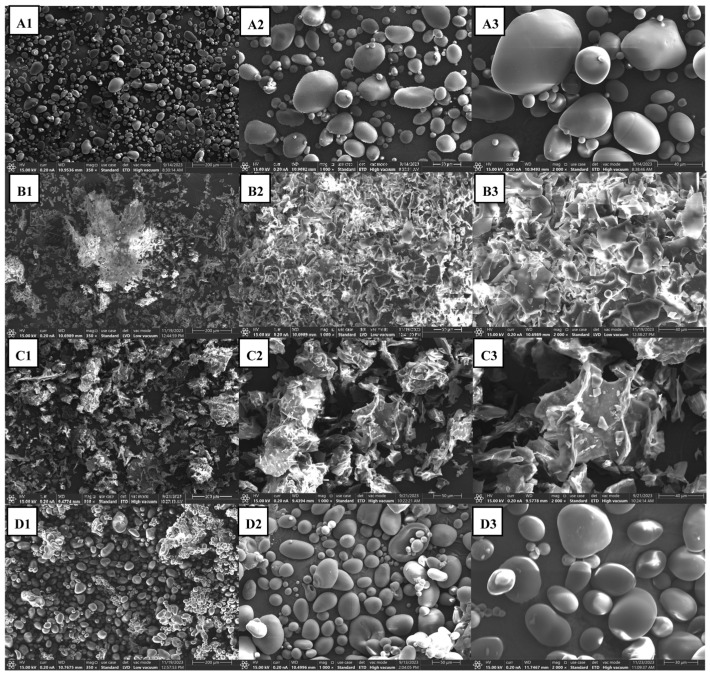
SEM micrographs of (**A**) smooth roundish granules of native potato starch at magnifications of (**A1**) ×350, (**A2**) ×1000, (**A3**) ×2000; (**B**) flaked starch after autoclave processing followed by a retrogradation at 40 °C for 24, at magnifications of (**B1**) ×350, (**B2**) ×1000, (**B3**) ×2000; (**C**) flaked starch after autoclave processing followed by a retrogradation at 4 °C for 24, at magnifications of (**C1**) ×350, (**C2**) ×1000, (**C3**) ×2000; (**D**) defected roundish starch granules after HHP processing followed by a retrogradation at 4 °C for 24 at magnifications of (**D1**) ×350, (**D2**) ×1000, (**D3**) ×2000.

**Figure 3 foods-14-00880-f003:**
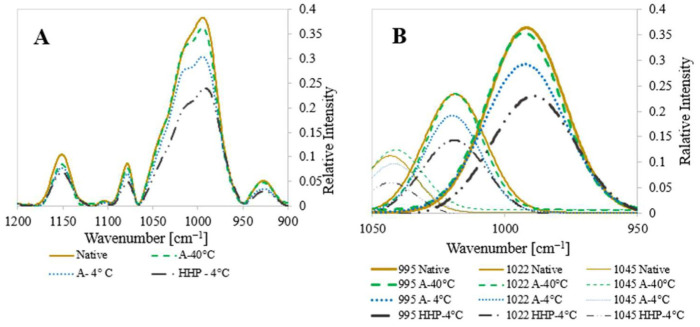
(**A**) FTIR spectrum between 900 and 1200 cm^−1^ of native starch (**˗˗˗˗˗**) and processed potato starch autoclaved and retrograded at 40 °C (**˗ ˗ ˗ ˗ ˗**); autoclaved and retrograded at 4 °C (**․․․․․**); and processed by HHP and retrogradation at 4 °C (▬ ▪ ▪ ▬). (**B**) Deconvoluted spectrum for the peaks at 995 cm^−1^, 1022 cm^−1^, and 1045 cm^−1^.

**Figure 4 foods-14-00880-f004:**
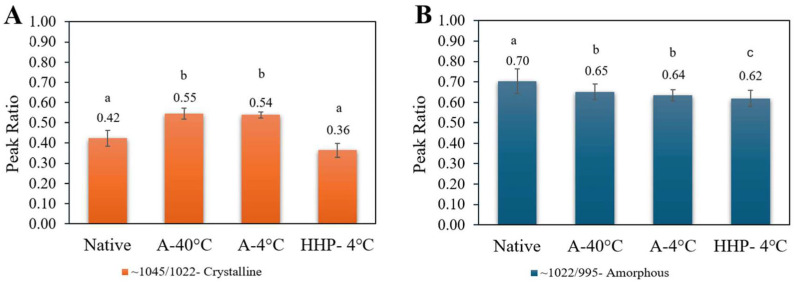
FTIR peak ratios indicative of (**A**) amorphous (wavenumber 1022/995R) regions and (**B**) crystalline regions (wavenumber 1045/1022R) of native starch, A-40 °C, A-4 °C, and HHP-A-4 °C. Statistically significant values are denoted by letter where *p* < 0.05 (n = 9).

**Figure 5 foods-14-00880-f005:**
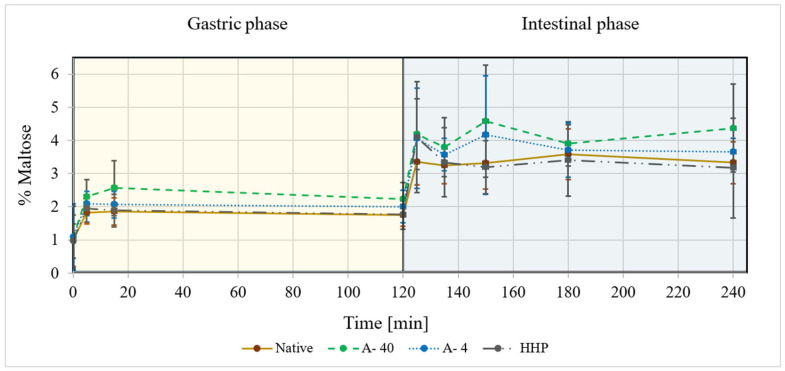
The in vitro digestion of starch expressed in terms of %maltose released as a function of time during gastric and small-intestinal digestion. For native potato starch (Native **˗˗˗˗˗**); autoclaved and retrograded at 40 °C (A-40 **˗ ˗ ˗ ˗ ˗**); autoclaved and retrograded at 4 °C (A-4 **․․․․․**); processed by HHP and retrogradation at 4 °C (HHP▬ ▪ ▪ ▬), according to an adapted protocol for semi-dynamic digestion based on the INFOGEST protocol. (n = 3).

**Figure 6 foods-14-00880-f006:**
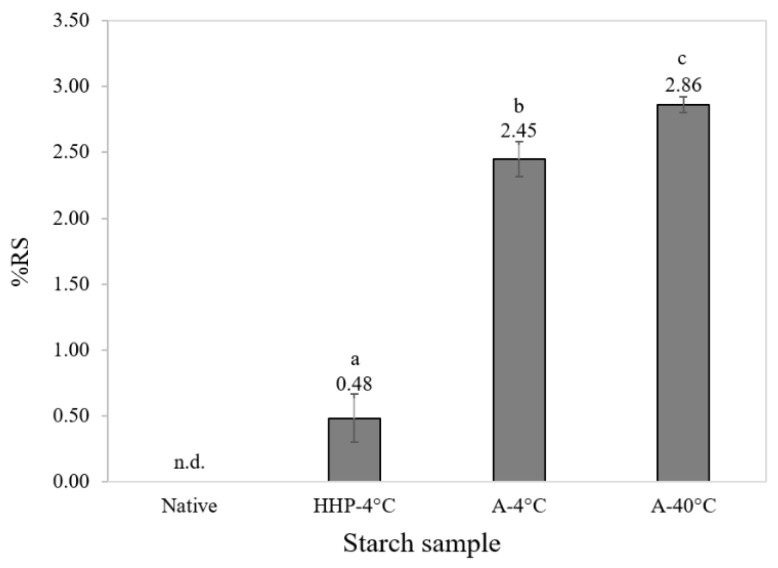
%RS (*w*/*w*) content obtained from Native potato starch, potato starch autoclaved and retrograded at 40 °C potato starch autoclaved and retrograded at 4 °C, and potato starch processed by HHP and retrogradation at 4 °C. Statistically different values denoted with letters where *p* < 0.005 (n = 4).

**Figure 7 foods-14-00880-f007:**
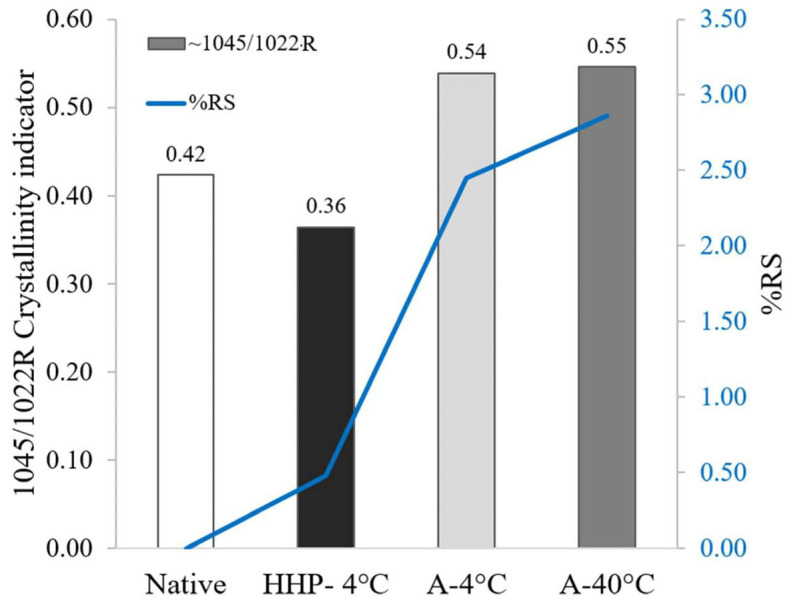
Concurrence between FTIR indicator of starch crystallinity (ratio of wavenumbers 1045/1022R) and %RS obtained for Native potato starch, potato starch autoclaved and retrograded at 40 °C, potato starch autoclaved and retrograded at 4 °C, and potato starch processed by HHP and retrogradation at 4 °C.

**Figure 8 foods-14-00880-f008:**
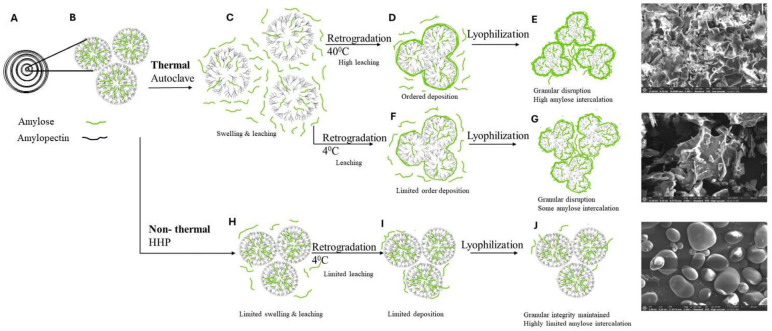
Illustration of the changes occurring during starch processing. (A) Starch granule concentric ring model, (B) starch granules composed of amylose and amylopectin, (C) during autoclaving starch granules swell and amylose chains leach to the bulk, (D) during retrogradation at 40 °C amylose chains are orderly deposited on the disrupted granules. Some amylose chains remain free in the bulk; (E) after lyophilizing, amylose chains that were in the bulk are intercalated between the disrupted granules; (F) during retrogradation at 4 °C amylose chains are less mobile and are deposited on the starch granules in a less orderly manner; (G) after lyophilizing, amylose chains remaining in the bulk are intercalated between to the disrupted granules; (H) during HHP, the structure of starch granules remains mostly intact, with limited leaching of amylose chains; (I) during retrogradation of HHP process starch at 4 °C fewer amylose chains (compared to thermally processed starch) are deposited on the starch granules; (J) after lyophilization, low levels of amylose chains present in the bulk become loosely attached to the granules.

## Data Availability

The data presented in this study are available on request from the corresponding author due to the size, diversity and complexity of the data.

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
