# Peer review of "Potential of Process-Induced Modification of Potato Starch to Modulate Starch Digestibility and Levels of Resistant Starch Type III"

_foods, 2025, doi:10.3390/foods14050880_

Round 1

Reviewer 1 Report

Comments and Suggestions for Authors

This study investigated the impact of processing methods like autoclaving and high hydrostatic pressure (HHP) on potato starch digestibility and RS content, followed by retrogradation at different temperatures. The research employed several methods to characterize structural changes in starch samples using X-ray diffraction (XRD), attenuated total reflectance-Fourier transform infrared (ATR-FTIR) spectroscopy, and scanning electron microscopy (SEM). It is a very interesting study, especially figure 8 illustrated the changes in starch structure during starch processing.

1) The authors want to increase the content of resistant starch type III in the modified starch products. Although RS classification has been influenced by many factors, more characteristic parameters of RS type 3 should be introduced in the introduction section and the results section.

2) Line 103-110, the moisture content in starch sample before and after autoclaving treatment should be given.

3) Line 89, which microorganism was amyloglucosidase (10113) from ?

4) In figure 7, the icon for 1045/1022R should be same for four samples.

5) Line 82-89, the authors give two types of alpha-amylase, respectively from Aspergillus oryzae (86250) and Bacillus licheformis (A4551). In Line 178 and Line 216, each alpha-amylase should be noted. 

6) In the conclusion section, the further study for the present study should be given.

Author Response

Detailed response to reviewers’ comments

We sincerely thank the reviewers for their time and building criticisms towards refining this manuscript and reinforcing its clarity and legibility. We have carefully revised the manuscript, re-edited some of the figures and elaborated the introduction, discussion as well as the conclusion sections. Below are our detailed responses.

Reviewer#1

This study investigated the impact of processing methods like autoclaving and high hydrostatic pressure (HHP) on potato starch digestibility and RS content, followed by retrogradation at different temperatures. The research employed several methods to characterize structural changes in starch samples using X-ray diffraction (XRD), attenuated total reflectance-Fourier transform infrared (ATR-FTIR) spectroscopy, and scanning electron microscopy (SEM). It is a very interesting study, especially figure 8 illustrated the changes in starch structure during starch processing.

1) The authors want to increase the content of resistant starch type III in the modified starch products. Although RS classification has been influenced by many factors, more characteristic parameters of RS type 3 should be introduced in the introduction section and the results section.

Among the parameters influencing the increase of RS III are the processing method (autoclave, heat moisture, etc. ) and processing cycling, as well as the conditions during processing, such as heating temperature and duration which are critical for maximizing RSIII yield. When addressing retrogradation, the increase in RS III can be influenced by parameters such as duration and temperature.

  • Discussion on RS has been elaborated in the revised manuscript.

2) Line 103-110, the moisture content in starch sample before and after autoclaving treatment should be given.

  • Updated in the revised manuscript.

3) Line 89, which microorganism was amyloglucosidase (10113) from ?

  • Enzyme produced in Aspergillus Niger. This has been clarified in the revised manuscript.

4) In figure 7, the icon for 1045/1022R should be same for four samples.

  • Corrected in the revised manuscript.

5) Line 82-89, the authors give two types of alpha-amylase, respectively from Aspergillus oryzae (86250) and Bacillus licheformis (A4551). In Line 178 and Line 216, each alpha-amylase should be noted. 

  • Updated in the revised manuscript.

6) In the conclusion section, the further study for the present study should be given.

  • A comment about future research added to the revised manuscript.

Reviewer 2 Report

Comments and Suggestions for Authors

The manuscript presents important information regarding starch modification and the importance of RS.

Some aspects should be emphasized, which are detailed below:

11.     The authors could briefly include the methods and processes that allow starch modification. Which ones are frequently used, what findings are there.

22.     In 3.1. The authors could comment on why there are these differences in the FTIR analysis and what is represented in Figure 4, from the point of view of starch structures.

33.     In 3.2. What is the effect of applying heat on starches? Since HHP does not present a significant difference with native starch. What happens structurally with RS? Figure 8 is very explanatory, but the authors could kindly indicate the modifications of starches.

4.4     The authors should indicate how RS was calculated in the methodology.

Author Response

The manuscript presents important information regarding starch modification and the importance of RS.

Some aspects should be emphasized, which are detailed below:

  1. The authors could briefly include the methods and processes that allow starch modification. Which ones are frequently used, what findings are there.
  • A section on chemical and physical modifications along with relevant references have been added into the introduction of the revised manuscript.
  1. In 3.1. The authors could comment on why there are these differences in the FTIR analysis and what is represented in Figure 4, from the point of view of starch structures.
  • Autoclaving and retrogradation appear to enhance short-range crystallinity, resulting in an increased ratio of crystallinity, as represented by the 1045/1022R peak. This is discussed in the revised manuscript.
  1. In 3.2. What is the effect of applying heat on starches? Since HHP does not present a significant difference with native starch. What happens structurally with RS? Figure 8 is very explanatory, but the authors could kindly indicate the modifications of starches.
  • While heat increases the mobility of amylose chains, thereby enhancing their leaching from the starch granule, HPP did not appear to have this effect. This is discussed in the revised manuscript.
  1. The authors should indicate how RS was calculated in the methodology.
  • A comment clarifying RS yield was calculated from the ratio of the sediment obtained after the digestion to the total starch subjected to digestion has been added to Section 2.6.

Reviewer 3 Report

Comments and Suggestions for Authors

Q1: Why autoclave (121 °C, 20 min) and autoclave (550 MPa, 15 min)? The selection basis is not fully explained and lacks literature support.

Q2: What are the potential differences between semi-dynamic in vitro and static digestion selected?

Q3: The article mentions that "HHP treatment did not significantly change the crystallinity of starch", but the 5.6° peak was missing in all treatment groups in the XRD plot, does this phenomenon need to be further explained?

Q4: In the analysis of SEM, no relevant references were cited. The results of the analysis are simpler.

Q5: The legend of the digestion curve in Figure 5 is inconsistent with the description (HHP is abbreviated as “HPP” in the legend)

Q6: Some DOI links are missing, do they need to be supplemented?

Author Response

Q1: Why autoclave (121 °C, 20 min) and autoclave (550 MPa, 15 min)? The selection basis is not fully explained and lacks literature support.

  • Conditions selected to simulate commercially viable conditions. This is emphasized in the revised manuscript.

Q2: What are the potential differences between semi-dynamic in vitro and static digestion selected?

  • As described in Duijsens et al, 2022, semi dynamic models offer better insight into digestion kinetics as well as improved bio-relevance. This is now emphasized in the revised manuscript.

Q3: The article mentions that "HHP treatment did not significantly change the crystallinity of starch", but the 5.6° peak was missing in all treatment groups in the XRD plot, does this phenomenon need to be further explained?

  • As described, HHP samples did not appear to be significantly divergent than those of native starch. This is also affirmed by SEM and discussed in the revised manuscript.

Q4: In the analysis of SEM, no relevant references were cited. The results of the analysis are simpler.

  • Indeed, SEM images offer “simpler” analyses and we focused on discussing their relevance to the other results of the study to offer a careful discussion and avoid overinterpretations.

Q5: The legend of the digestion curve in Figure 5 is inconsistent with the description (HHP is abbreviated as “HPP” in the legend)

  • Typo corrected in the revised manuscript.

Q6: Some DOI links are missing, do they need to be supplemented?

  • We have made efforts to check style and grammar to improve legibility as well as references, using Mendeley references manager.

Reviewer 4 Report

Comments and Suggestions for Authors

I believe that the article is very interesting by it subject. The use of HPP is of a high interest for readers and producers. Some remarks: In the introduction part the authors affirm (line 74-77) that more recent studies...my question is how recent? what studies. The authors must cite the sources. 

Materials and methods (lines 111-121) - why authors used HPP conditions such as pressure 550 MPa and time 15 minute? Please explain. Also what are the devices used for the fallowing steps - line 117 (lyophillization,  pulverization, e.g. )

The part of results and discussion is vague presented. I am wonder if authors had presented separated these parts what they will write to the discussion? They only presents XRD results, SEM and the others with few discussions. For example FTIR has 4 peaks. What represent each peak? No discussion related on chemical composition and how it affect FTIR spectra is provided. 

I really think that to publish an article in a such a prestigious journal as Food more discussions are necessary. If not, the chance that this article to be cited is very low. 

Author Response

I believe that the article is very interesting by it subject. The use of HPP is of a high interest for readers and producers. Some remarks: In the introduction part the authors affirm (line 74-77) that more recent studies...my question is how recent? what studies. The authors must cite the sources. 

  • References and citations have been updated in the revised manuscript.

Materials and methods (lines 111-121) - why authors used HPP conditions such as pressure 550 MPa and time 15 minute? Please explain.

  • Conditions selected to simulate commercially viable conditions.

Also what are the devices used for the fallowing steps - line 117 (lyophillization,  pulverization, e.g. )

  • Clarified in the revised manuscript.

The part of results and discussion is vague presented. I am wonder if authors had presented separated these parts what they will write to the discussion? They only presents XRD results, SEM and the others with few discussions. For example FTIR has 4 peaks. What represent each peak? No discussion related on chemical composition and how it affect FTIR spectra is provided. 

  • As also noted by other reviewers, the methods deployed have been in routine use in starch research, hence, we focused our discussion on the new insights that can be gained, in the prospect of the study’s goals.

I really think that to publish an article in a such a prestigious journal as Food more discussions are necessary. If not, the chance that this article to be cited is very low.

  • The novelty of the work has been emphasized in the revised manuscript. Notably, we believe that the evidence linking processing, crystallinity and starch architectures to digestibility is of great interest these days. Moreover, Figure 8 will be of high value to food professionals.

Reviewer 5 Report

Comments and Suggestions for Authors

This study provides an in-depth investigation into the effects of processing techniques on potato starch digestibility and resistant starch (RS) content. The authors employ a combination of structural characterization techniques—X-ray diffraction, attenuated total reflectance-Fourier transform infrared spectroscopy, and scanning electron microscopy—to examine how autoclaving and high hydrostatic pressure followed by retrogradation, influence starch structure and enzymatic digestion. The findings contribute valuable knowledge to the field of food science, particularly in improving starch-based food products for better health outcomes.

However, certain aspects require further clarification.

Major

1. Can the Authors clarify the novelty of your work in the context of existing research on starch modification? How does your study differentiate itself from previous investigations in terms of methodology, findings, or applications? Additionally, what specific advancements or new insights does your research contribute to the understanding of starch digestibility and resistant starch enhancement?

2. How does the cost-effectiveness of the starch modification methods used in your study compare to already existing techniques? Are there significant economic or scalability advantages of using autoclaving and high hydrostatic pressure (HHP) over traditional methods such? How do the outcomes of your study (in terms of resistant starch content and digestibility) compare to previously established methods in both efficiency and practicality for large-scale food production?

3. The study compares autoclaving and HHP but does not fully justify why these specific methods were chosen over others.

4. The bibliography in the study is not sufficiently up-to-date, as 41 out of the 58 cited references are older than five years. Given the rapid advancements in research, it is essential that the authors provide the reader with the most recent findings.

Minor

1. The manuscript exhibits some inconsistencies in formatting, including punctuation errors and a somewhat disorganized bibliography.

Author Response

This study provides an in-depth investigation into the effects of processing techniques on potato starch digestibility and resistant starch (RS) content. The authors employ a combination of structural characterization techniques—X-ray diffraction, attenuated total reflectance-Fourier transform infrared spectroscopy, and scanning electron microscopy—to examine how autoclaving and high hydrostatic pressure followed by retrogradation, influence starch structure and enzymatic digestion. The findings contribute valuable knowledge to the field of food science, particularly in improving starch-based food products for better health outcomes.

However, certain aspects require further clarification.

Major

  1. Can the Authors clarify the novelty of your work in the context of existing research on starch modification? How does your study differentiate itself from previous investigations in terms of methodology, findings, or applications? Additionally, what specific advancements or new insights does your research contribute to the understanding of starch digestibility and resistant starch enhancement?
  • We have made an effort to clarify the work’s novelty. The present study uses an up to date in vitro digestion system, following the consensus INFOGEST digestion protocol for food research. Moreover, we offer novel insights into the complex interplay between thermal and non-thermal processing methods combined with retrogradation on starch digestion in a holistic approach looking into RDS, SDS and RS. Lastly, Figure 8 illustrates our underlying hypothesis to help professionals better understand the impact of processing on starch digestion dynamics.
  1. How does the cost-effectiveness of the starch modification methods used in your study compare to already existing techniques? Are there significant economic or scalability advantages of using autoclaving and high hydrostatic pressure (HHP) over traditional methods such? How do the outcomes of your study (in terms of resistant starch content and digestibility) compare to previously established methods in both efficiency and practicality for large-scale food production?
  • This academic work sought to underpin the link between thermal and non-thermal processing combined with retrogradation on starch properties and the ramifications to starch digestion using the most biorelevant digestion protocols. It may offer food manufacturers with opportunities to process starches into “healthier” starch variants.
  1. The study compares autoclaving and HHP but does not fully justify why these specific methods were chosen over others.
  • This academic work sought to underpin the link between thermal and non-thermal processing combined with retrogradation based on a literature review on physical methods to modify starches. This notion is now better explained in the revised manuscript and supported by additional references.

  1. The bibliography in the study is not sufficiently up-to-date, as 41 out of the 58 cited references are older than five years. Given the rapid advancements in research, it is essential that the authors provide the reader with the most recent findings.
  • The references have been updated in the revised manuscript. It contains a mix of seminal work in the field of starch research along side recent advancements and discoveries.

Minor

  1. The manuscript exhibits some inconsistencies in formatting, including punctuation errors and a somewhat disorganized bibliography.

We have made efforts to check style and grammar to improve legibility as well as references, using Mendeley references manager.

Round 2

Reviewer 4 Report

Comments and Suggestions for Authors

The authors made a minimum revision. They did not revised at all the part of results and discussion. If they made a separation according to the journal template of these parts I am not sure that they can write very much to the discussion part. They say that discussion is focus on the new insights that can be gained but the results obtained do not have been explained? Is really strange for me.  

Reviewer 5 Report

Comments and Suggestions for Authors

The Authors have updated their manuscript, however, two major objections from the previous review remain unaddressed.

  1. The question regarding how the cost-effectiveness of the starch modification methods used in this study compares to existing techniques has been completely omitted. A proper discussion on this aspect is essential to highlight the practical relevance of the study.
  2. Response to Q3 simply states that one of the chosen method is thermal and the other non-thermal. However, this does not explain why these specific methods were chosen. Why was autoclaving selected as the thermal method and HPP as the non-thermal method? Why were other potential methods not considered?

Further, could the Authors specify which references have been added in response to the previous review comments. A clear indication of new citations will help assess the revisions more effectively.
